# A Psychosocial Genomics Pilot Study in Oncology for Verifying Clinical, Inflammatory and Psychological Effects of Mind-Body Transformations-Therapy (MBT-T) in Breast Cancer Patients: Preliminary Results

**DOI:** 10.3390/jcm10010136

**Published:** 2021-01-03

**Authors:** Mauro Cozzolino, Stefania Cocco, Michela Piezzo, Giovanna Celia, Susan Costantini, Valentina Abate, Francesca Capone, Daniela Barberio, Laura Girelli, Elisa Cavicchiolo, Paolo Antonio Ascierto, Gabriele Madonna, Alfredo Budillon, Michelino De Laurentiis

**Affiliations:** 1Department of Human, Philosophical and Educational Sciences, University of Salerno, 84084 Fisciano (SA), Italy; mcozzolino@unisa.it (M.C.); giovanna.celia@libero.it (G.C.); lgirelli@unisa.it (L.G.); ecavicchiolo@unisa.it (E.C.); 2Department of Breast and Thoracic Oncology, Division of Breast Medical Oncology, Istituto Nazionale Tumori IRCCS “Fondazione G. Pascale”, 80131 Naples, Italy; s.cocco@breastunit.org (S.C.); m.piezzo@breastunit.org (M.P.); 3Experimental Pharmacology Unit—Mercogliano Laboratory, Istituto Nazionale Tumori IRCCS “Fondazione G. Pascale”, 80131 Naples, Italy; s.costantini@istitutotumori.na.it (S.C.); f.capone@istitutotumori.na.it (F.C.); a.budillon@istitutotumori.na.it (A.B.); 4Psychology Unit, Istituto Nazionale Tumori IRCCS “Fondazione G. Pascale”, 80131 Naples, Italy; vale.abate@gmail.com (V.A.); danielabarberio68@gmail.com (D.B.); 5Department Melanoma, Cancer Immunotherapy and Development Therapeutics, Istituto Nazionale Tumori IRCCS “Fondazione G. Pascale”, 80131 Naples, Italy; p.ascierto@istitutotumori.na.it (P.A.A.); g.madonna@istitutotumori.na.it (G.M.)

**Keywords:** breast cancer, mind-body transformations therapy (MBT-T), inflammation, cytokines

## Abstract

Several studies have highlighted the key role of chronic inflammation in breast cancer development, progression, metastasis, and therapeutic outcome. These processes are mediated through a variety of cytokines and hormones that exert their biological actions either locally or distantly via systemic circulation. Recent findings suggest that positive psychosocial experiences, including psychotherapeutic interventions and therapeutic mind-body protocols, can modulate the inflammatory response by reducing the expression of genes/proteins associated with inflammation and stress-related pathways. Our preliminary results indicate that a specific mind-body therapy (MBT-T) could induce a significant reduction of the release of different cytokines and chemokines, such as SCGFβ, SDF-1α, MCP3, GROα, LIF, and IL-18, in the sera of breast cancer patients compared to a control group, suggesting that MBT-T could represent a promising approach to improve the wellness and outcome of breast cancer patients.

## 1. Introduction

### 1.1. Breast Cancer, Tumor Microenvironment and Inflammation

Breast cancer (BC) is the most common malignant neoplasm and the second most common cause of cancer-related death in women, with 276,480 new cases estimated in 2020 in the USA, representing about 30% of all diagnosed tumors. Despite the high morbidity of the disease, BC hasa better prognosis when compared to other aggressive tumors. According to the American Cancer Society’s biennial update on female BC statistics, the overall five-year survival rate is 99% for those patients with localized disease and 86% for those patients with regional disease and drops to 27% for those patients with distant metastases. It is estimated that 20–30% of early stage breast cancers (eBC) will develop metastatic disease, while 6–10% of all diagnosed women have stage IV disease at diagnosis [1,2]. Due to the high heterogeneity, both at a cellular and at a molecular level, BC comprises several differenttumor subtypes. According to gene expression profiles, BC subtypes can be classified into at least three subtypes: luminal tumors, which are positive for estrogen and/or progesterone receptors (ER/PgR); HER2-enriched, which overexpress the ERBB2 oncogene (HER2); and basal like, also known as triple negative tumors (TNBC), which lack hormone receptors and HER2 amplification [3]. Each subtype has different characteristics, in terms of incidence, response to treatment, risk of disease progression, and sites of metastases [4,5]. The presence/absence of these receptors is used to guide the therapeutic choices, such as hormonal therapy, chemotherapy, immunotherapy and other targeted therapies, in both the early and advanced/metastatic settings [6,7,8,9]. Recurrence and metastases are the main contributors for the treatment failure and poor outcomes of BC patients; therefore the identification of patients with an unfavorable prognosis is critical in the formulation of optimal, individualized, and multimodal therapeutic strategies [10]. In recent years, the acquisition of new knowledge regarding molecular characteristics of BC and the peritumoral microenvironment suggest that BC consist not only of neoplastic cells, but also of significant alterations in tumor microenvironment, made up of multiple cell types (e.g., fibroblasts, leukocytes, adipocytes), extracellular matrix (ECM), physical properties (e.g., PH, oxygen content), and soluble factors, such as cytokines, hormones, growth factors, and enzymes [11,12,13,14,15].

In this scenario, where cancerous cells and microenvironment are co-protagonists, the inflammatory process has to be considered a crucial mechanism for the recurrence and metastasis [16,17,18,19,20]. Inflammation is a physiological process in response to acute tissue damage resulting from multiple causes such as ischemic damages, infections, exposure to toxins, chemical irritation and/or different types of trauma [21,22]. When the inflammatory stimulus persists, the inflammation becomes chronic. In the inflamed site a complex signaling network is created, involving a large number of growth factors, cytokines, different types of leukocytes, lymphocytes, other inflammatory cells, and chemokines. Chronic inflammation is involved in all phases of tumor development: initiation, progression and metastasis. Initially, inflammation plays a role in tumor suppression, stimulating an antitumor immune response, afterwards it seems to stimulate tumor growth [16,23]. The intensity, duration and nature of inflammation may explain this apparent contradiction. A key aspect of tumor microenvironment is the cytokine-mediated communication between tumor and peritumoral cells, where cytokines and chemokines show many activities that allow cell-cell communication [24]. One of the main differences between normal cells and tumor cells is represented by the continuous proliferation of the latter, which soon results in a deficiency of nutrients and oxygen; the state of hypoxia created during tumor growth induces many cytokines and chemokines [25,26]. Since cytokines production is a highly complex and multifactorial mechanism, the identification of a specific role in the pathogenesis of disease of a single cytokine is difficult and likely useless, while could be very useful exploring the complex network of interactions utilized to regulate both the cytokines synthesis and the similar receptors. The complexity of the cytokine system and their interaction, also known as cytokinome, can be explored by analyzing cytokine panels simultaneously, in order to provide an overview of the antagonistic and synergistic interactions among different cytokines, which involve many and often redundant pathways [27,28].

### 1.2. Novel Multidisciplinary Approaches for BC Treatment and Follow-Up

It has been well known that genes/proteins can interact with the environment to modulate behavior and cognition under conditions of disease and health [29]. These interactions involve a particular class of genes, frequently defined as activity genes or experience-dependent genes, which can be activated by signals from the physical and psychosocial environment modulating complex physiological and psychological functions of the organism [30,31,32,33,34,35]. Psychosocial stressors appear to have dynamic experience-dependent effects on gene expression through the involvement of numerous interrelated circuits which mediate the effects of psychosocial stress on physiology, cell biology and finally on gene expression [36]. Recent evidences suggest that positive psychosocial experiences, including psychotherapeutic interventions and therapeutic mind-body protocols, can modify the transcriptional dynamics of leukocytes under pathological conditions related to stress such as chronic disease, cancer and psychiatric disorders, by reducing the expression of genes associated with inflammatory response and stress-related pathways, and improve the mind-body health through proper negotiation of pathways to stress response [37,38,39,40,41].

The transition from healthy woman to BC patient is generally associated with significant physical, psychological, and social challenges. In these women psychological stress strongly increases when confronting a BC diagnosis and undergoing cancer treatment. These elevated stress levels can in turn heighten inflammation and seem to be maintained during the post-treatment period in breast survivors as well [42,43,44,45]. For this reason, the development of multidisciplinary evidence-based care during the post-treatment phase is a key area of cancer research. Over the last few years, the psychosocial and cultural origin of chronic stress has been investigated as risk factor for BC development and recurrence, as well as the unmet needs of BC survivors [37,41,46,47] leading to development of novel therapeutic approaches, based on integration of medical and psychological needs [37,48].

Several studies have elucidated the role of neuroendocrine regulation of downstream physiological and biological pathways relevant to cancer development, also demonstrating how subjective stressful experiences, defined in the literature as bio-behavioral factors, may influence tumor growth and progression, via sympathetic nervous system (SNS) and hypothalamic- pituitary-adrenal axis (HPA) activation [37]. Also, the inflammatory process has been recently associated to neoplasm transformation and tumor growth. Specifically, the NF-κB mediated signal transduction is implicated in the regulation of viral replication, autoimmune diseases, inflammatory response, tumorigenesis and apoptosis.

In this context, the development of new technologies, such as the emerging area of omics sciences, and the mind-body medicine are changing the way of approaching BC, taking into account also a holistic perspective, in order to integrate mind and body within a single comprehensive framework [49,50,51,52]. This kind of intervention, aimed to reduce symptoms related to stress diseases, can affect relevant processes related to cancer growth and progression by reducing the inflammation status and augmenting the immune response [35,38,48,53]. Functional genomics studies show interesting connections between mind-body therapies and immune system. In particular, recent studies have shown that mind-body therapies are able to generate an overall reduction of the expression of factors related to inflammatory response, such as NF-κB, and to regulate numerous pathways involved in apoptosis and cell proliferation [37,41,47,48,54].

In cancer patients the effects of these interventions are able to counter processes related to cancer growth through the reduction of inflammation and the increase of immune response [48]. Mind-body therapies are able to reduce stress, eliciting the relaxation response (RR), which in turn modulates genes expression. Different studies have shown a reduced expression of genes associated with the inflammatory response and the pathways related to stress [41], as well as they demonstrate that a good Stress Management is able to reduce the expression of genes related to the biochemical inflammatory response in women with breast cancer [37]. Indeed, mind-body therapies have been shown to improve quality of life and to increase the chance of survival for patients [47,53,54,55]. This evidence highlights the importance of integrating traditional procedures with mind-body clinical protocols which take into account physical, psychological and genomic aspects of oncologic disease at the same time [54]. Although there is a growing body of literature in this direction, it is necessary to expand the knowledge in this field though clinical trials.

### 1.3. Mind-Body Therapeutic Approach

Psychosocial genomics is a new top-down approach (from mind to body) that examines the modulation of gene expression in response to psychological, social, and cultural experiences [32,38,51,56,57,58,59,60]. Several studies have shown that the experience of novel environmental situations promotes activity- and experience-dependent gene expression, brain plasticity, anti-inflammatory response, and stem cell healing processes, as well as the genome capacity to quickly respond to individuals’ psychosocial experiences [38,49,50,51,61,62,63,64,65]. 

The psychosocial genomics theoretical paradigm encompasses a clinical method called mind-body transformations therapy (MBT-T), which is a therapeutic integrated approach grounded in studies by Erickson and Rossi’s mind-body therapy and applies for both groups and individuals [51,66,67]. It moves from a naturalistic perspective of therapy and it is based on the use of individual’s natural biological rhythms to set the best conditions for activating inner mind-body healing processes, in order to face the different challenges of individual’s organism, such as stress-related dysfunctions of different psychological and mind-body chronic disorders [35,50,51,57,65]. MBT-T has proved to be a valid method for promoting well-being and to reduce stress [68,69,70,71,72]. Specifically, different studies have highlighted that MBT-T can modulate experience dependent genes to reduce symptoms of the stress related disorders and to facilitate mind-body healing [35,38,58,68,69,70].

## 2. Results and Discussion

### 2.1. Decrease of Cytokine Concentrations in Breast Cancer Patients after MBT-Treatment

Cytokines and chemokines are small proteins that play a role in cell-to-cell communication by paracrine or autocrine signaling. They were thought to induce immune responses to foreign threats or inflammation and play prominent roles in human biology and diseases [73]. Several studies have showed that the dynamics of cross talk between the immune system and cancer cells mediated by cytokines and chemokines changes during cancer initiation, progression, and therapeutic interventions [74,75,76]. Indeed, tumor-secreted cytokines and chemokines play a key role in shaping the tumor microenvironment and promote metastasis by facilitating tumor dissemination, motility and invasiveness in breast cancer [74,75,76,77]. Therefore, we decided to evaluate the cytokines profiling in the sera of breast cancer patients enrolled in control and experimental arms at baseline (T0), 1 h after the first MBT-T treatment (T1), and the end of the MBT-T treatment (Tf). In particular, the release of SCGFβ, SDF-1a, MCP3, and IL-18 was significantly reduced in sera of patients collected after just 1 MBT-T(T1) compared to the control group (non-treated patients) (Figure 1A,B), while the reduction of MCP3, GROa and LIF was observed at the end of treatment (EOT) (Figure 1C,D). All these significant cytokines, found to be decreased in the MBT-T-treated group, are known as to be pro-inflammatory. This confirms that their decrease can be considered as an index of an anti-inflammatory effect of the MBT-T treatment. In particular, stem cell growth factor β (SCGF) is a member of the C-type lectin superfamily acting as a growth factor for primitive hematopoietic progenitor cells. It has been shown to be involved in the survival of cancer stem cells, and, consequently, implicated in the malignant progression in cancer. Its high levels were also found in the circulating tumor cells (CTCs) derived from breast cancer patients [78]. IL-18 is an interferon-gamma-inducing factor that acts on both Th1 and Th2 inflammatory responses and is involved in several cancers, where correlate with development of chemo-resistance, high stage and mortality [79]. In breast cancer, it is associated with cancer promotion and progression since serum levels were significantly higher in breast cancer patients than in healthy subjects, and in metastatic patients compared to non-metastatic [80,81]. Recently, high levels of IL-18 were correlated with a poor prognosis in surgically treated breast cancer patients. In particular, relapse-free survival (RFS) was found to be significantly worse in IL-18-high patients than in those with IL-18 low levels [82]. Gro-a (CXCL1) is a chemokine associated to cancer growth and proliferation, angiogenesis and metastasis [83] which up-expression was found in patients with different cancers. In particular, in BC patients it is correlated with tumor grade, disease recurrence and decreased survival [84]. Some findings showed a role of Gro-α in promoting cellular migration and invasion, epithelial-mesenchymal transition (EMT) and BC metastasis via NF-kB/SOX4 activation [85]. It is highly expressed in ER-negative subtype, where can stimulate cell migration and invasion via the ERK/MMP2/9 pathway [86]. 

SDF-1α (CXCL12) plays a key role in cancer chemotaxis through its CXCR4 receptor. The expression of chemokine SDF-1a is associated with increased invasion, mammosphere formation, metastasis, chemoresistance, angiogenesis and a poor prognosis in breast cancer [87,88]. Other findings highlighted that it result over-expressed in invasive breast carcinoma samples compared with normal adjacent tissues, and in advanced and metastatic tumors [89]. MCP-3 (monocyte chemotactic protein-3) is a chemokine that plays a pivotal role in tumorigenesis, promoting cancer progression by supporting the formation of the tumor microenvironment and facilitating tumor invasion and metastasis. In particular, MCP-3 resulted to be up-expressed in the plasma of breast cancers patients compared to the healthy controls, and in patients with poorly differentiated carcinomas [90]. Moreover, high mRNA levels of MCP-3 were related to decreased overall survival and relapse-free survival [91]. LIF (Leukemia inhibitory factor) is a secreted cytokine involved in some biological processes including differentiation of leukemia cells, inflammatory response, neuronal development, and cancer progression by activating and regulating JAK/STAT3, AKT, EKR1/2 and mTOR signal pathways [92,93]. It’s up-expression has been verified in many cancers, including breast cancer, and associated with poor prognosis on recurrent free survival [94].

### 2.2. Changes in Well-Being, Anxiety and Depression (Measured through Psychological Scales)

Participants’ well-being, anxiety and depression were also evaluated through psychological scales from breast cancer patients enrolled in control and experimental arm, at baseline (T0) and at the end of the MBT-T treatment (Tf). Results of the 2 × 2 mixed-design factorial ANOVA showed not statistically significant interaction between time and condition in participants’ well-being, anxiety and depression (F(1,8) = 1.302; *p* = 0.287; F(1,12) = 1.948; *p* = 0.188; F(1,12) = 1.470; *p* = 0.249 respectively). Table 1 displays the means of well-being, depression, and anxiety. Even though the results showed not significant interactions, they appear to be in the expected direction: participants in the experimental condition showed an increase in well-being and a reduction in the level of anxiety through T0 and Tf as compared to participants in the control condition, who showed a reduction in well-being and an increase in anxiety and depression.

## 3. Conclusions

Current research in the psychosocial genomics of stress is achieving a quiet but significant evolution in understanding the fundamentals of the etiology and proliferation of a variety of human cancers. This research is a step forward in the theoretical and practical advances for the amelioration and cure of cancer. 

MBT-T integrates the main psychotherapeutic orientations and represents an innovative model of evidence-based psychotherapy able to promote well-being and to reduce depression and anxiety.

The main objective of this pilot study was to ascertain whether this specific mind-body therapeutic protocol (MBT-T) was able to modulate the expression of activity-experience dependent proteins decreasing the stress response, reducing the activation of inflammatory pathways, improving the quality of life and facilitating a better mind-body health response in BC patients who have completed loco-regional treatment and adjuvant chemotherapy.

Preliminary results of the investigation of circulating markers of inflammation indicate that MBT-T produced a significant reduction of several pro-inflammatory cytokines and chemokines involved in the mechanisms of drug resistance and cancer progression in breast cancer. The data show that the intervention has determined a modulation of inflammatory pathways in the acute and long-term therapy. This could prevent the risk of recurrence in patients with breast cancer. As regards the assessment of psychological variables, the exploratory results have shown an increase in well-being (measured with the well-being scale of the CORE-OM) and a reduction in the levels of depression and anxiety (measured with HADS scale) in patients in the experimental condition, as compared to participants in the control condition. Although these first results are not statistically significant, they suggest the importance of MBT-T in the improvement of the quality of life in BC patients. 

Despite these first encouraging results, the present pilot study has some limitations that should be considered when interpreting the results. As regards the psychological assessment, a limitation is represented by its relatively small sample size and the resultant lack of statistical power in detecting effects. Despite the difficulties in contacting such a specific population, it is worth conducting future longitudinal studies on larger samples, in order to explore the possible existence of important effects. Moreover, more aspects of psychological distress need to be assessed in order to better understand the general improvement in health and quality of life of BC patients. As regards the general results of the present study, one of the limitations is the not-large sample size of the patients involved, which was due to the difficulties in contacting and following over time this specific population. Our findings will hopefully be confirmed by future research that might involve a larger number of patients and more uniform cohorts. In this context, it could also be relevant to stratify breast cancer populations for adjuvant hormonal therapy, in order to evaluate any potential effect on stress and inflammation. Another limitation is that the study was carried out on patients with non-metastatic breast cancer. Future research should also consider other types of breast cancer and include patients with metastatic breast cancer. Finally, our study assessed the clinical, inflammatory and psychological effects of the MBT-T therapeutic approach that has proved to present certain advantages over traditional mind-body interventions. However, it will be necessary to apply other traditional techniques in order to compare their effectiveness in cancer patients. 

In spite of these limitations, the present study suggested that the mind-body therapeutic protocol implemented (MBT-T) might have a significant role in improving the quality of life and in facilitating a better mind-body health response in patients affected by cancer. 

Compared to other techniques, MBT-T presents many advantages: it is very easy to learn and can be performed in minutes (so it is not time-consuming), without the need of specific equipment, special premises or training, and body skills. MBT-T can be administered individually as well as to a large number of subjects at the same time and it is shown to be effective after just one session. Therefore, this therapeutic protocol represents a clinical opportunity in the treatment of diseases which are characterized by difficulties in their management and elevated healthcare costs, such as cancer. 

## 4. Materials and Methods 

### 4.1. Study Designand Patient Eligibility Criteria

A monocentric, interventional, non-pharmacological, open-label, randomized study on patients with non-metastatic breast cancer was carried out, and this involved a follow-up procedure to experimentally validate the genomic and epigenetic effects of the innovative mind-body technique (MBT-T). During first phase of our study, 28 patients were recruited and 5 dropped out during the study. The patients were enrolled at the end of the standard adjuvant treatments (surgery, chemotherapy, radiotherapy) after an interview with the reference oncologist and psychologist and were randomly assigned at a one-to-one ratio to the following arms: -Arm A: Standard follow-up (control arm): 7 patients undergoing the standard follow-up procedures as scheduled by reference oncologist-Arm B: Standard follow-up + MBT-T intervention (experimental arm): 16 patients undergoing the standard follow-up procedures as scheduled by reference oncologist, with the addition of a biweekly psychological treatment (MBT-T) lasting 4 months (8 sessions). Each therapy is organized in a 90–120 min’ group sessions (16 patients).-In both groups, blood samples (6–8 mL) have been collected before the treatment (T0), after 1 h of the first treatment (T1), after 2 months (T2), and at the end of treatment (Tf). Then, an additional blood sample has been collected after 2 months from Tf. Sera were collected by centrifugation (2250× g for 10 min at 4 °C), aliquoted, and stored at −80 °C until analysed, as previously described.

Written informed consent was obtained from all patients. The trial was conducted in accordance with the Good Clinical Practice guidelines and Declaration of Helsinki. The study protocol and any modifications were approved by an independent ethics committee and an institutional review board.

Main inclusion criteria are: diagnosis of invasive early stage breast cancer confirmed histologically without evidence of residual, locally relapsed or metastatic disease, and classified as a clinical stage from I (T1, N0, M0) to IIIC (any T, N3, M0); female; minimum level of education: high school diploma patients; completion of adjuvant therapy within 6 months of inclusion in the study.

Main exclusion criteria are: patient unable to read or understand study documents; patient with any significant disease or other clinical condition that, according to the judgement of the researcher, could interfere with the evaluation of the study (chronic inflammatory diseases, psychiatric syndromes, use of psycho-pharmaceuticals, etc.) or with appropriate participation in the study; metastatic disease.

### 4.2. Psychological Measures

In the present pilot study, we focused on the following measures: (1) the well-being scale of CORE-OM [95,96], a self-assessment scale that measures general psychological distress; (2) the Hospital Anxiety and Depression Scale (HADS) [97,98], a self-assessment scale developed to detect states of depression, anxiety and emotional distress. 

### 4.3. Primary and Secondary Endpoints

The main objective of this pilot study is to detect whether a specific mind-body therapeutic protocol (MBT-T) is able to modulate the expression of activity-experience dependent genes, decreasing the stress response, reducing the activation of inflammatory pathways, improving the quality of life, and facilitating a better mind-body health response in BC patients who have completed loco-regional treatment and adjuvant chemotherapy. 

The general objectives are:-To translate the most recent findings in the field of neuroscience, genomic research and mind-body medicine into cancer clinical practice through the conduct of a randomized clinical trial that aims to demonstrate the effectiveness and sustainability of a particular mind- body approach (mind-body transformations therapy—MBT-T) compared to traditional approaches.-To understand the determinants of the therapeutic outcome through the study of the genome that can clarify the molecular mechanisms underlying the clinical efficacy of our MBT-T approach on cancer patients.-To identify and validate genomic-based classifiers that act as potential predictors of the clinical benefit in order to promote the personalization and optimization of treatment.

### 4.4. Procedures

#### 4.4.1. Mind-Body Transformation Therapy (MBT-T) 

MBT-T follows a structured protocol, based on a 4-stage creative process and it is characterized by a very easy to learn procedure allowing individuals to obtain stress reduction without the need for traditional, complex, and intricate methods [68,69,70]. MBT-T protocol can be applied both to individuals and groups and it is based on the use of: the natural activity-rest physiologic cycle (BRAC) [59,99]; the ultradian biological rhythms; the biological plasticity showed by individuals genes expression; the relaxion response (RR) [31,38,49,51,58]. 

The clinical protocol foresees movements of the palms of the hands that the therapist shows to the patients and that the patient will perform in turn, within each of the four phases of the creative process: Phase 1—initiation and creative expectation (focus on the sensations experienced in hic et nunc); Phase 2—incubation and access to emerging experience (review of problems); Phase 3—illumination with a new perspective oriented to problem solving (insight and problem-solving); Phase 4—assessment, planning (reality assessment and self-care) [68]. This will allow to start a mind-body therapeutic dialogue, able to generate new awareness for the management, the facing, and/or the solution of the problems that the patient faces daily in the relationship with the disease. 

#### 4.4.2. Bioplex Assay 

This method was carried out according to the manufacturer’s instructions (Bio-Plex Bio-Rad) to assess the cytokines levels. The Bio-Plex Pro Human Cytokine 21-Plex Immunoassay (IL-1α, IL-2Rα, IL-3, IL-12 (p40), IL-16, IL-18, CTACK, GRO-α, HGF, IFN-α2, LIF, MCP-3, M-CSF, MIF, MIG, β-NGF, SCF, SCGF-β, SDF-1α, TNF-α, and TRAIL) has been used on sera of seven breast cancer patients included in the control arm and sixteen breast cancer patients, enrolled in the experimental arm (randomized MBT-T study during follow-up after curative treatments), before (T0), after 1 h of treatment (T1) and at the end of the treatment (Tf). Protein levels were determined using a Bio-Plex array reader (Luminex, Austin, TX, USA) that quantifies multiplex immunoassays in a 96-well format with very small fluid volumes. The analyte level was calculated using a standard curve, with software provided by the manufacturer (Bio-Plex Manager Software, Austin, TX, USA). 

### 4.5. Statistical Considerations

Group sample sizes of 45 and 45 achieve 80% power to detect a difference of 0.6 between the group means with estimated group standard deviations of 1.0 and 1.0 and with a significance level (alpha) of 0.05000 using a two-sided two-sample t-test. 

The concentrations of all the cytokines (expressed in pg/mL) evaluated in each group before treatment (T0) were compared with those at the end of first treatment (T1) and at the end of the treatment (Tf) using multivariate statistical analysis (Partial least squares discriminant analysis—PLS-DA). Variable Important Plot (VIP) was used to identify the more modulated cytokines during treatment and/or time. Red or green boxes indicated if the cytokine levels increased or decreased during treatment, respectively.

As regards the psychological scales, in order to evaluate whether patients belonging to the experimental group had better therapeutic outcome with reference to quality of life, anxiety and depression compared to the control group, a series of two-factor mixed-design analysis of variance (ANOVA) were conducted using time as within-subjects factor (baseline vs. end of treatment) and conditions as between-subjects factor (experimental vs. control). The 2 × 2 mixed-design ANOVA was conducted separately for each psychological variable of the study, using the IBM SPSS Statistics for Windows, version 23 (IBM Corp., Armonk, NY, USA).

### 4.6. Protection of Trial Participants

The participants taking part in the trial are not at risk to their health. The testing protocol has been developed with a careful assessment of the risk/benefit ratio, which should not be biased in favor of risk. No injury to the physical integrity of the patient is expected except the blood collection indicated in the different stages of the trial (tests that are not essential for the therapeutic or diagnostic purpose of the study will not be conducted). The confidentiality of personal and clinical data will be guaranteed, in compliance with the privacy legislation with the acceptance and signature of the participants as well as a consent form of information on the trial. In this way, the patient can exercise the right to information and the right to self-determination for participation in the experimental study and the correct use of the situation of the control group.

The study was approved by the Ethics Committee of Istituto Nazionale Tumori IRCCS Fondazione “G. Pascale” of Naples Italy, with registration number 11/17, resolution 460 of 21 June 2017.

## Figures and Tables

**Figure 1 jcm-10-00136-f001:**
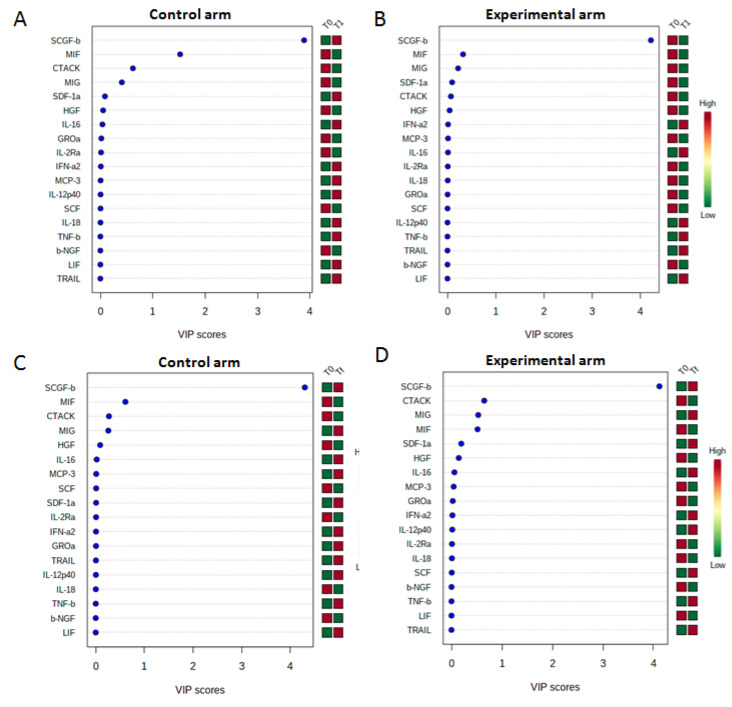
Partial least squares discriminant analysis (PLS-DA) was performed to compare the cytokine concentrations on sera from Breast Cancer Patients enrolled in the randomized Mind-Body Transformations Therapy (MBT-T) study. In particular, we show variable important plot (VIP) related to the comparison between cytokines concentrations evaluated by Bio-Plex Pro Human Cytokine 21-Plex Immunoassay on sera from 7 and 16 Breast Cancer Patients enrolled in control and experimental arm, respectively, at baseline (T0) and after 1 h of the first treatment (T1) (**A**,**B**) and at baseline (T0) and the end of the MBT-T treatment (Tf) (**C**,**D**). The shown cytokines are interleukin-1α (IL-1α), interleukin-2Rα (IL-2Rα), interleukin-3 (IL-3), interleukin-12 (IL-12), interleukin-16 (IL-16), interleukin-18 (IL-18), cutaneous T cell-attracting chemokine (CTACK), Growth-regulated protein alpha (GRO-α), hepatocyte growth factor (HGF), interferon-α2 (IFN-α2), Leukemia inhibitory factor (LIF), Monocyte chemotactic protein-3 (MCP-3), Macrophage migration inhibitory factor (MIF), monokine induced by gamma interferon (MIG), beta nerve growth factor (β-NGF), colony-stimulating factor (SCF), stem cell growth factor-beta (SCGF-β), stromal cell-derived factor 1alpha (SDF-1α), tumor necrosis factor-beta (TNF-β) and Tumor necrosis factor (TNF)-Related Apoptosis Inducing Ligand (TRAIL).

**Table 1 jcm-10-00136-t001:** Means (and standard deviation) of the psychological measures of the study for both arms (16 Experimental patients vs. 7 Control patients) and time (Baseline vs. End of treatment).

Constructs	Experimental Arm		Control Arm	
	Baseline	End of treatment	Baseline	End of treatment
Well-being	5.00 (1.54)	6.50 (1.87)	5.50 (3.08)	4.0 (3.75)
Depression	11.75 (0.78)	11.85 (0.68)	12.83 (0.90)	14.50 (0.78)
Anxiety	11.00 (0.76)	9.75 (0.99)	10.00 (0.88)	10.66 (1.14)

## Data Availability

The data presented in this study are available on request from the corresponding author.

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
