# Peer review of "A Psychosocial Genomics Pilot Study in Oncology for Verifying Clinical, Inflammatory and Psychological Effects of Mind-Body Transformations-Therapy (MBT-T) in Breast Cancer Patients: Preliminary Results"

_jcm, 2021, doi:10.3390/jcm10010136_

Round 1

Reviewer 1 Report

Cozzolino et al. investigated the impact of specific mind-body therapy (MBT-T) on cytokine and chemokine release in sera of breast cancer patients. The topic of this manuscript is interesting. However, I have some concerns and suggestions.

Please provide the raw data of the multiplex cytokine measurements.

Line 47 and 185: please rephrase “expression” of cytokines as they are if any released from cells into the blood stream.

Line 67: please add a reference.

Line 181: please add a reference.

Line 185 and Figure 1: When was the sample T1 in the control group collected? Is this the same timeframe? In the Material and Method section it is mentioned that additional blood samples were collected at T2. I can not find any analysis of these samples in the manuscript.

Line 186: abbreviations must be defined when first mentioned in the text.

Figure legend 1: Please indicate the n-numbers. Moreover, please define: VIP, MBT, cytokine profiling (à multiplex), all cytokine abbreviations, the timepoint T1 in the control arm and the used statistical analysis.

Table 1: What kind of statistical analysis was performed? Please indicate the n-numbers. Please make the terms “experimental” or “intervention” consistent.

Line 241-244: Which three constructs do the authors talk about? And how can this be in line with the hypothesis, if the statistical analysis does not strengthen that MBT treatment is helpful for breast cancer patients?

Line 279 – 282: From my point of view this is an overstatement as the authors stated in Line 238-240 that “ANOVA showed not statistically significant interaction between time and condition in participants’ well-being, anxiety and depression (F(1,8) = 1.302; p = .287; F(1,12) = 1.948; p = .188; F(1,12) = 1.470; p = .249 respectively)”.

Line 298-303: Why did the authors choose only 7 patients undergoing normal follow-up procedures versus 16 patients with MBT therapy? Would it not be better to have a more balanced group distribution?

Author Response

Cozzolino et al. investigated the impact of specific mind-body therapy (MBT-T) on cytokine and chemokine release in sera of breast cancer patients. The topic of this manuscript is interesting. However, I have some concerns and suggestions.

Thank you very much for your appreciative comment concerning our study. Thank you also for your useful suggestions. We have revised the whole manuscript in order to address each of them.

Please provide the raw data of the multiplex cytokine measurements.

Ok, you can find the raw data in attach.

Line 47 and 185: please rephrase “expression” of cytokines as they are if any released from cells into the blood stream.

We have appropriately changed “expression” with “release”

Line 67: please add a reference.

1 reference was added as you suggested.

Line 181: please add a reference.

3 references were added as you suggested.

Line 185 and Figure 1: When was the sample T1 in the control group collected? Is this the same timeframe? In the Material and Method section it is mentioned that additional blood samples were collected at T2. I can not find any analysis of these samples in the manuscript.

In this preliminary phase of the study we supposed to analyze just more relevant time points for each patient (T0, T1 and Tf). A comprehensive analysis of all time points will be performed on achievement of complete enrolment of 90 patients.

Line 186: abbreviations must be defined when first mentioned in the text.

Thank you for your comment. We have reported in the text the correct abbreviation (MBT-T) that refers to the particular methodology applied in the present study and that it has been already used in the text.

Figure legend 1: Please indicate the n-numbers. Moreover, please define: VIP, MBT, cytokine profiling (à multiplex), all cytokine abbreviations, the timepoint T1 in the control arm and the used statistical analysis.

Thank you for your comment. We rewrote the figure legend 1 inserting the details suggested by the reviewer 1.

Table 1: What kind of statistical analysis was performed? Please indicate the n-numbers. Please make the terms “experimental” or “intervention” consistent.

Thank you for your comment. ANOVA analysis was performed. We added the numbers of the patients and modified “intervention” with “experimental. In the text we have also clarified the contents of Table 1.

Line 241-244: Which three constructs do the authors talk about? And how can this be in line with the hypothesis, if the statistical analysis does not strengthen that MBT treatment is helpful for breast cancer patients?

Thank you very much for your comment, we have rephrased this part in order to clarify it further, as follow: “Even though the results showed not significant interactions, they were in the expected direction:  participants in the experimental condition showed an increase in well-being and a reduction in the level of anxiety through T0 and Tf as compared to participants in the control condition, who showed a reduction in well-being and an increase in anxiety and depression.”

Line 279 – 282: From my point of view this is an overstatement as the authors stated in Line 238-240 that “ANOVA showed not statistically significant interaction between time and condition in participants’ well-being, anxiety and depression (F(1,8) = 1.302; p = .287; F(1,12) = 1.948; p = .188; F(1,12) = 1.470; p = .249 respectively)”.

Thanks for your comment. We have revised this statement in order to address your concern, as follow: “In spite of these limitations, the present study suggested that the mind-body therapeutic protocol implemented (MBT-T) might have a significant role in improving the quality of life and in facilitating a better mind-body health response in patients affected by cancer.”

Line 298-303: Why did the authors choose only 7 patients undergoing normal follow-up procedures versus 16 patients with MBT therapy? Would it not be better to have a more balanced group distribution?

We are fully agree with this remark, indeed the study expect to enrol 90 patients equally randomized in the two arms (45 control vs 45 MBT). Unfortunately, during this preliminary phase of the study, 5 control patients dropped out and they have been excluded by the analysis. 

Reviewer 2 Report

Dear authors,

The manuscript entitled “A psychosocial genomics pilot study in oncology for verifying clinical, inflammatory and psychological effects of mind-body transformations-therapy (MBT-4 T) in breast cancer patients: preliminary results” is reviewed.

Cozzolino  et al has done excellent pilot study for the assessment of clinical, inflammatory and psychological effects of MBT-T in patients with breast cancer. The study would provide a reference data in breast cancer and might be a good fit for the scope of the journal. However, study subjects are few therefore, results should be verified in a large cohort. The study performed on non-metastatic breast cancer patients. Other types of breast cancer patients like metastatic should be included to verify the current results using this treatment. The results should be shown using other traditional techniques for the comparison.

Author Response

Dear authors,

The manuscript entitled “A psychosocial genomics pilot study in oncology for verifying clinical, inflammatory and psychological effects of mind-body transformations-therapy (MBT-4 T) in breast cancer patients: preliminary results” is reviewed.

Cozzolino  et al has done excellent pilot study for the assessment of clinical, inflammatory and psychological effects of MBT-T in patients with breast cancer. The study would provide a reference data in breast cancer and might be a good fit for the scope of the journal.

We would like to thank the Reviewer for his/her appreciative comments and all his/her suggestions. We have revised our manuscript accordingly, with the purpose of following his/her recommendations.

However, study subjects are few therefore, results should be verified in a large cohort. The study performed on non-metastatic breast cancer patients. Other types of breast cancer patients like metastatic should be included to verify the current results using this treatment. The results should be shown using other traditional techniques for the comparison.

We think that the reviewer has raised an important point and we have expanded the limitations, which refers to possible future research, accordingly.

Reviewer 3 Report

This manuscript describes a pilot study into the realm of mind-body transformations-therapy (MBT-T) and impact it may have on inflammatory markers in breast cancer patients.  Although interesting, the robust statistical confirmation is not there and clearly requires additional study with more uniform cohorts and larger numbers of patients.  In addition there are a few queries:

  1. Did you control between the two arms of the study for:
    1. use of not of adjuvant hormonal therapy
    2. use of antibiotics or not for an ongoing UTI for example
    3. presence or not of an inflammatory illness such as asthma and active status and concomitant treatment or not for this
  2. Need clarification regarding the inclusion criteria (lines 311-319); are metastatic patients (M disease) included or not?
  3. In the Conclusions I would substitute evolution in place of breakthrough (line 251) and delete profound (line 252)

The bibliography is comprehensive.  There are a moderate amount of stylistic and grammatical modifications that I would suggest, some of which are noted:

  • line 40 Several studies
  • line 44 can modulate the inflammatory response
  • line 48 in the sera
  • line 49 a control group, suggesting that
  • line 56 in the USA representing
  • line 60 for those patients
  • line 63 both at a cellular and at a molecular
  • line 67 which lack hormone receptors and HER2 amplification.
  • line 71 both the early and advanced/metastatic settings.
  • line 71 metastases are the
  • line 75 signaling network is created
  • line 94 which soon results in a deficiency 
  • line 99 the complexity of the cytokine system
  • line 104 can interact with the environment
  • line 119 when confronting a BC diagnosis
  • lines 120-121 and seem to be maintained during the ....survivors as well.
  • line 129 defined in the literature as bio-behavorial factors
  • line 131 Also, the inflammatory process has been recently
  • line 136 medicine are changing the way of approaching BC
  • line 140 and augmenting the immune response
  • line 151 have been shown 
  • line 152. to improve quality of life
  • line 179 between the immune 
  • line 184 control and experimental arms at baseline
  • line 187 (non-treated patients)
  • line 191 In particular, stem cell
  • line 197 it is associated with cancer promotion
  • line 198 serum levels were significantly
  • line 199 subjects, and in metastatic
  • line 200 IL-18 were correlated with a poor prognosis
  • line 201 (RFS) was found to be

Author Response

This manuscript describes a pilot study into the realm of mind-body transformations-therapy (MBT-T) and impact it may have on inflammatory markers in breast cancer patients.  

We are very grateful to the Reviewer for all his/her appreciative comments about our research and for his/her useful suggestions regarding the manuscript. Please see below our responses to your concerns.

Although interesting, the robust statistical confirmation is not there and clearly requires additional study with more uniform cohorts and larger numbers of patients. 

Thank you for your comment. In the revised version of the manuscript we have expanded the limitations in order to include the Reviewer’s point.

In addition there are a few queries:

  1. Did you control between the two arms of the study for:
    1. use of not of adjuvant hormonal therapy

since the low sample size of this preliminary analysis, we didn’t stratify the patients for hormonal therapy. We plan to perform this analysis on the achievement of complete intention to treat population.

    1. use of antibiotics or not for an ongoing UTI for example

as per exclusion criterion #2 “Patient with any disease or other significant clinical condition that, in the investigator's judgment, could interfere with the study assessments (chronic inflammatory diseases, psychiatric syndromes, use of psychotropic drugs, etc.) or with adequate participation in the study” patients with UTI or under antibiotics treatment have been excluded.

    1. presence or not of an inflammatory illness such as asthma and active status and concomitant treatment or not for this

as reported before, patients with inflammatory illness have been excluded.

  1. Need clarification regarding the inclusion criteria (lines 311-319); are metastatic patients (M disease) included or not?

As per exclusion criterion #3, patients with metastatic disease have been excluded.

  1. In the Conclusions I would substitute evolution in place of breakthrough (line 251) and delete profound (line 252)

These sentences have been changed as you suggested.

The bibliography is comprehensive.  

Thank you for your appreciation.

There are a moderate amount of stylistic and grammatical modifications that I would suggest, some of which are noted:

Thank you for your indications. We have carefully revised the manuscript accordingly.

  • line 40 Several studies
  • line 44 can modulate the inflammatory response
  • line 48 in the sera
  • line 49 a control group, suggesting that
  • line 56 in the USA representing
  • line 60 for those patients
  • line 63 both at a cellular and at a molecular
  • line 67 which lack hormone receptors and HER2 amplification.
  • line 71 both the early and advanced/metastatic settings.
  • line 71 metastases are the
  • line 75 signaling network is created
  • line 94 which soon results in a deficiency 
  • line 99 the complexity of the cytokine system
  • line 104 can interact with the environment
  • line 119 when confronting a BC diagnosis
  • lines 120-121 and seem to be maintained during the ....survivors as well.
  • line 129 defined in the literature as bio-behavorial factors
  • line 131 Also, the inflammatory process has been recently
  • line 136 medicine are changing the way of approaching BC
  • line 140 and augmenting the immune response
  • line 151 have been shown 
  • line 152. to improve quality of life
  • line 179 between the immune 
  • line 184 control and experimental arms at baseline
  • line 187 (non-treated patients)
  • line 191 In particular, stem cell
  • line 197 it is associated with cancer promotion
  • line 198 serum levels were significantly
  • line 199 subjects, and in metastatic
  • line 200 IL-18 were correlated with a poor prognosis
  • line 201 (RFS) was found to be

Round 2

Reviewer 1 Report

The authors have responded to each comment/suggestion and have improved the quality of the manuscript.

Author Response

The authors have responded to each comment/suggestion and have improved the quality of the manuscript.

Thank you for your comments.

Reviewer 3 Report

This manuscript describes a pilot study into the realm of mind-body transformations-therapy (MBT-T) and impact it may have on inflammatory markers in breast cancer patients.  The revised manuscript is improved. The abstract better describes the preliminary, although interesting, nature of the research. There still is a moderate degree of repetitiveness and too many run on sentences.

Given the prevalence of adjuvant hormonal therapy in breast cancer management, I believe you should note the fact that you did not stratify for this in this preliminary study and plan to do so with subsequent studies.

A few additional grammatical/stylistic suggestions:

line 74: the identification of patients with an unfavorable prognosis is critical in the formulation of optimal, individualized and multimodal therapeutic strategies.

line 147: In cancer patients... are able to counter processes related to cancer growth through

line 183: play a key role in shaping the tumor microenvironment...and invasiveness in breast cancer

line 185: to evaluate the cytokine profile in the sera of breast cancer patients

line 191: significant cytokines, found to be decreased in the MBT-T-treated

line 192: as an index of

line 337: without evidence of residual, locally relapsed or metastatic disease, and classified...

Author Response

Given the prevalence of adjuvant hormonal therapy in breast cancer management, I believe you should note the fact that you did not stratify for this in this preliminary study and plan to do so with subsequent studies.

Thank you for your comment. We have inserted this consideration in conclusions, lines 323-324.

A few additional grammatical/stylistic suggestions:

we have updated the manuscripy as you suggested.

line 74: the identification of patients with an unfavorable prognosis is critical in the formulation of optimal, individualized and multimodal therapeutic strategies.

line 147: In cancer patients... are able to counter processes related to cancer growth through

line 183: play a key role in shaping the tumor microenvironment...and invasiveness in breast cancer

line 185: to evaluate the cytokine profile in the sera of breast cancer patients

line 191: significant cytokines, found to be decreased in the MBT-T-treated

line 192: as an index of

line 337: without evidence of residual, locally relapsed or metastatic disease, and classified...